Manipulation of light quality is an effective tool to regulate photosynthetic capacity and fruit antioxidant properties of Solanum lycopersicum L. cv. ‘Microtom’ in a controlled environment

Vitale Ermenegilda 1
Velikova Violeta 2
Tsonev Tsonko 3
Costanzo Giulia 1
Paradiso Roberta 4
http://orcid.org/0000-0002-9718-2941 Arena Carmen 1 5 c.arena@unina.it
1 Department of Biology, University of Naples Federico II , Naples , Italy
2 Institute of Plant Physiology and Genetics, Bulgarian Academy of Sciences , Sofia , Bulgaria
3 Institute of Biophysics and Biomedical Engineering, Bulgarian Academy of Sciences , Sofia , Bulgaria
4 Department of Agricultural Sciences, University of Naples Federico II , Portici , Italy
5 BAT Center-Center for Studies on Bioinspired Agro-Environmental Technology , Portici , Italy
Levizou Efi
Electronic publication date: 2022 Jul 1
Publication date: 2022
Volume: 10
Electronic Location ID: e13677
Received 2022 Feb 24; Accepted 2022 Jun 13
Copyright: © 2022 Vitale et al.
Copyright year: 2022
Copyright holder: Vitale et al.
License: This is an open access article distributed under the terms of the Creative Commons Attribution License, which permits unrestricted use, distribution, reproduction and adaptation in any medium and for any purpose provided that it is properly attributed. For attribution, the original author(s), title, publication source (PeerJ) and either DOI or URL of the article must be cited.
License URL: https://creativecommons.org/licenses/by/4.0/

Keywords: Photosystem II, Gas exchanges, Leaf functional traits, Light quality, Photochemistry, Rubisco, Antioxidant production, Tomato

Funding: The authors received no funding for this work.

==============================
Light quality plays an essential role in setting plant structural and functional traits, including antioxidant compounds. This paper aimed to assess how manipulating the light spectrum during growth may regulate the photosynthetic activity and fruit bioactive compound synthesis in Solanum lycopersicum L. cv. ‘Microtom’ to improve plant physiological performance and fruit nutritional value. Plants were cultivated under three light quality regimes: red-green-blue LEDs (RGB), red-blue LEDs (RB) and white fluorescent lamps (FL), from sowing to fruit ripening. Leaf functional traits, photosynthetic efficiency, Rubisco and D1 protein expression, and antioxidant production in fruits were analyzed. Compared to FL, RGB and RB regimes reduced height and increased leaf number and specific leaf area, enhancing plant dwarf growth. The RGB regime improved photosynthesis and stomatal conductance despite lower biomass, favoring Rubisco synthesis and carboxylation rate than RB and FL regimes. The RB light produced plants with fewer flowers and fruits with a lower ascorbic acid amount but the highest polyphenol content, antioxidant capacity and SOD and CAT activities. Our data indicate that the high percentage of the green wavelength in the RGB regime promoted photosynthesis and reduced plant reproductive capacity compared to FL and RB. Conversely, the RB regime was the best in favoring the production of health-promoting compounds in tomato berries.

Introduction

The demand for healthy fresh food has increased according to the global population rise in the last decades. However, satisfying this need has led to the intensification of non-sustainable agriculture practice and the overuse of broad cultivation areas, with consequent overexploitation of natural resources (FAO, 2017). Furthermore, the open field cultures are increasingly threatened by the risks and uncertainties associated with biotic and abiotic stresses, such as pest attacks, drought, and frost, exacerbated by the ongoing climate change (Pandey et al., 2017), compelling need for new cultivation approaches (Dutta-Gupta, 2017; FAO, 2019). The Controlled Environment Agriculture (CEA) has emerged as a feasible alternative, as it optimizes the plant growth environment by minimizing the interactions with the external factors (Dutta-Gupta, 2017; Amitrano et al., 2018; Pennisi et al., 2019). The manipulation of light quality in CEA through light-emitting diodes (LEDs) technology modifies plant morphological, anatomical, and physiological traits (Arena et al., 2016; Yang et al., 2017), allowing to select the more appropriate light regime to improve crop productivity and food quality for a specific crop. This approach is worthy of attention to reduce the overuse of resources needed for massive crop production and plant cultivation in extreme environments such as hot and cold deserts or extraterrestrial platforms. In the view of space colonization, in the last decade, the National Aeronautics and Space Administration (NASA) has strongly encouraged the development of CEA and LED-based plant growth systems on the International Space Station (ISS) to support the realization of future colonies on the Moon and Mars (Massa et al., 2006; Wheeler & Morrow, 2010; Gomez & Izzo, 2018).

Changes in growth and photosynthesis induced by different light wavelengths are strictly linked to species, but some evidence iswidely recognized. Generally, red and blue wavelengths are most efficiently utilized for photosynthesis and influence the synthesis of PSII D1 protein and Rubisco (Kato et al., 2015; Izzo et al., 2020; Vitale et al., 2020). More specifically, red light influences the photosynthetic apparatus development, biomass accumulation, and stem elongation (Urbonavičiūte et al., 2007; Wang et al., 2009), and the level of soluble sugars (Cui et al., 2009) and fruits antioxidant compounds, like carotenoids and phenols (Panjai et al., 2017). Blue light is mainly involved in vegetative growth regulation, early photomorphogenesis, and stomata control (Chen et al., 2014; Singh et al., 2015; Izzo et al., 2020). A high proportion of blue wavelengths within the light spectrum, being more energetic, may cause light avoidance phenomena in chloroplasts, reducing photosynthesis (Loreto, Tsonev & Centritto, 2009; Pallozzi et al., 2013) and increasing the antioxidant production (i.e., lettuce, spinach) (Lester, 2006; Ohashi-Kaneko et al., 2007; Hasan et al., 2017) and protein biosynthesis (Li & Pan, 1994; Hasan et al., 2017) in some leafy vegetables. Finally, green light also plays a fundamental role in plant growth and development, involving seed germination and plant flowering (Wang & Folta, 2013), and modulation of fruits and sprouts (Samuolienė et al., 2011; Bantis, Ouzounis & Radoglou, 2016). In addition, the green wavelengths, penetrating deeply in the leaf mesophyll and lower canopy layers, promote photosynthesis and carbon gain in the deepest chloroplasts and inner canopy (Terashima et al., 2009; Smith, Mc Ausland & Murchie, 2017). Besides the traditionally used red and blue lights, green and orange enhance photosynthesis and translocation of assimilates by affecting source/sink relationships among plants. Green and orange bands improve the water use efficiency and promote plant growth through the accumulation of photoassimilates in leaves. This encourages to include the green light, besides red and blue, to project lighting systems in a growth-controlled environment (Lanoue, Leonardos & Grodzinski, 2018). Moreover, the green light strongly influences plant growth by acting on cryptochrome. Indeed, the green wavelengths may reverse cryptochrome blue-light mediated signals, such as dry biomass accumulation, stem growth inhibition, and anthocyanin production (Bouly et al., 2007; Zhang & Folta, 2012; Kusuma, Swan & Bugbee, 2021). Based on this evidence, the modulation of light spectral composition may be a practical approach for sustainable agriculture to obtain crops with specific characteristics in CEA and indoor cultivation.

The manipulation of the light spectrum to modulate photosynthesis and bioactive compound production still represents an open study field because light treatments promoting plant growth could be inappropriate for enhancing nutraceutical quality.

This study aimed to evaluate the effects of three different light quality regimes, white fluorescent (FL), red-green-blue (RGB), and red-blue (RB) LEDs light on growth, photosynthetic performance, and fruit antioxidant properties of Solanum lycopersicum L. cv. ‘Microtom’ plants.

Specific attention was devoted to the photosynthetic regulation in response to the different light quality treatments to assess the mechanisms allowing plants to improve productivity. To this purpose, gas exchanges, chlorophyll fluorescence measurements, chlorophyll and carotenoid content, and the expression of PSII D1 protein and Rubisco have been assessed.

The cultivar ‘Microtom’ was chosen in our experiment for a series of characteristics, such as short life cycle, compact size, fast growth, which makes it ideal for cultivation in small volumes at high plants density, compared to other tomato landraces (Scott & Harbaugh, 1989; Okazaki & Ezura, 2009; Saito et al., 2011; Shikata et al., 2016; Samuolienė et al., 2021).

The best light quality regime may be utilized to obtain cropswith enhanced productivity and high content of antioxidants, in specific indoor cultivation environments such as Space greenhouses or planetary platforms for providing fresh food to the crew (Colla et al., 2007; Saito et al., 2011; De Micco et al., 2014; Arena et al., 2019).

Materials and Methods

Plant material and growth conditions

Seeds of Solanum lycopersicum L. cv. ‘Microtom’, provided by Holland Online Vof (Amsterdam, The Netherlands), were sown in 3.0 L pots filled with peat soil and placed at 10–15 cm from each other (Scott & Harbaugh, 1989). Plants were cultivated in a climatized chamber under three different light regimes (five plants per treatment): white fluorescent light (FL) obtained by using fluorescent tubes (Lumilux L360W/640 and L360W/830, Osram, Germany); red-green-blue (RGB) and red-blue (RB) supplied by light-emitting diodes (LEDs) (LedMarket Ltd., Plovdiv, Bulgaria) with the following emission peaks: 630 nm red, 510 nm green, 440 nm blue. The used LEDs have some proportion of the adjacent to red, green and blue colors of the visible spectrum (Fig. 1) but for convenience we conditionallyaccept the designations RGB and RB meaning the peak wavelengths. An SR-3000A spectroradiometer was used to measure the spectral composition of the three light regimes (Fig. 1) with 10 nm resolution (Macam Photometrics Ltd., Livingston, Scotland, U.K.). Plant growth was followed from sowing to fruit ripening up to 100 DAS (days after sowing) under the following environmental conditions: photosynthetic photon flux density (PPFD) 300 ± 5 μmol photons m−2 s−1 for each light treatment, day/night air temperature 24/18 °C, relative air humidity 60–70%, photoperiod of 12 h. Plants were irrigated to pot capacity with tap water at a 2-day interval to reintegrate the water loss for evapotranspiration. Every 2 weeks, plants were fertilized with Hoagland’s solution.

Figure 1 Spectral distributions in the relative energy of FL (white fluorescent light), RB (red-blue) and RGB (red-green-blue) treatments.

Spectral distributions in the relative energy of the white fluorescent tubes and LEDs panels recorded for FL (white fluorescent light), RB (red-blue) and RGB (red-green-blue) treatments at the top of the plant canopy.

Measurements of plant growth and leaf functional traits

Plant growth measurements were carried out at 100 DAS. We considered: plant height (cm, considering the main stem), leaf number, fruit number, fruit weight (g FW per plant), epigeal plant biomass (EB, g FW per plant) as well as the ratios leaf biomass/epigeal biomass (LB/EB) and fruit biomass/epigeal biomass (FB/EB), where the epigeal biomass corresponds to the whole above-ground biomass. The flower number was monitored starting from 40 up to 70 DAS until the first fruits’ appearance, considering for each plant the sum of flowers measured within the range 40–70 DAS.

The determination of leaf functional traits (leaf area, LA; specific leaf area, SLA; leaf dry matter content, LDMC; relative water content, RWC), were assessed at 50 DAS on fully expanded leaves, according to methods reported in Cornelissen et al. (2003). LA (cm2) was measured by acquiring digital images and using ImageJ 1.45 program (Image Analysis Software, NIH, Bethesda, MD, USA). SLA was determined as the ratio between leaf area and dry leaf mass and expressed in cm2 g−1. LDMC was calculated as dry leaf mass to saturated fresh mass and reported in g g−1. RWC was expressed as a percentage of the ratio (fresh leaf mass – dry leaf mass)/(saturated leaf fresh mass – dry leaf mass). The saturated fresh mass was obtained by submerging the petiole of leaf blades in distilled water for 48 h in the dark at 15 °C, whereas the dry mass was determined after oven-drying leaves at 75 °C for 48 h.

Measurements of plant growth and leaf functional traits were determined on five plants for each light regime, collecting five leaves (one leaf per plant).

Gas exchange and chlorophyll a fluorescence measurements

Gas exchange and chlorophyllafluorescence measurements were carried out at 50 DAS on five plants per light regime. We selected one fully expanded leaf foreach plant to obtain five replicates per light treatment. The net CO2 assimilation (AN) and the stomatal conductance (gs) were measured using a portable leaf gas exchange system (LCpro+; ADC BioScientific, Hoddesdon, UK). The central leaflet of each compound leaf (5th from the stem base) was clamped into the gas exchange system cuvette (6.25 cm2) for measurements at 1,000 μmol photons m−2s−1 PPFD, 25 ± 2 °C leaf temperature and 50–60% relative humidity. The gas exchange measurements were conducted under red+10% blue light by means of light source of the gas exchange system (LCpro+; ADC BioScientific, Hoddesdon, UK). The photosynthesis and the stomatal conductance were calculated as indicated in Von Caemmerer & Farquhar (1981). The mesophyll conductance to CO2 diffusion (gm) was determined using the variable J method (Loreto et al., 1992), whereas the maximum rate of Rubisco carboxylation (Vcmax) was estimated as proposed by Farquhar, von Caemmerer & Berry (1980).

After gas exchange measurements, on the same leaves, chlorophyll a fluorescence was assessed by a fluorescence Monitoring System (FMS, Hansatech Instruments, King’Lynn, UK). The background fluorescence signal, Fo, was induced on 20 min dark-adapted leaves, by an inner light of about 2–3 μmol photons m−2 s−1, at a frequency of 0.5 kHz. Previous experiments demonstrated that 20 min are sufficient to obtain complete re-oxidation of PSII reaction centers (Shahzad et al., 2020). The maximum fluorescence level (Fm) in the dark-adapted state was determined with a 1 s saturating light pulse of about 6,000 μmol photons m−2s−1. The maximum PSII photochemical efficiency (Fv/Fm) was calculated as (Fm − F0)/Fm. Under illumination at plant growth irradiance (PPFD of 300 μmol photons m−2 s−1), the steady-state fluorescence (Fs) was measured, and maximum fluorescence (Fm’) in the light-adapted state was determined by applying a saturating pulse of 0.8 s with over 6,000 μmol photons m−2s−1. The quantum yield of PSII electron transport (ΦPSII) was calculated as (Fm’ − Fs)/Fm’ according to Genty, Briantais & Baker (1989), while the non-photochemical quenching (NPQ) was expressed as (Fm − Fm’)/Fm’ as reported in Bilger & Björkman (1990).

Photosynthetic proteins D1 and Rubisco and pigments content

After chlorophyll fluorescence and gas exchange measurements, the same leaves were collected to perform the protein extraction following the procedure of Wang et al. (2006) modified by Arena et al. (2019). Protein extracts were quantified with the Bradford assay Bradford (1976) and subjected to an SDS-PAGE (12%). The Western Blot analysis started treating the leaf samples with a blocking solution (100 mM Tris-HCl, pH 8.0, 150 mM NaCl, 0.1% Tween20, 10% Milk). In order to reveal the selected proteins, samples were then incubated with the respective primary and secondary antibodies (Agrisera, Vännäs, Sweeden): anti-PsbA (chicken, 1:5,000 v/v) for D1 protein, anti-RbcL (rabbit, 1:10,000 v/v) for Rubisco, anti-ACT (rabbit, 1:5,000 v/v) for Actin. Immuno-revelation was carried out using the kit for chemiluminescence (ECL Western Blotting Analysis System, Ge Healthcare, Chicago, IL, USA) by the Chemidoc system (Bio-Rad Laboratories, Hercules, CA, USA). The software Quantity One (Bio-Rad, Hercules, CA, USA) was used for the densitometric analysis to obtain quantitative information associated with the individual protein bands. The protein actin was used as loading control. The value of each band was normalized to the corresponding actin band. For all treatments, the density value was expressed in arbitrary units and represented as a bar diagram corresponding to the pixel volume of the protein band.

The photosynthetic pigments content, namely total chlorophylls (a+b) and carotenoids (x+c), were quantified on leaf samples of known area treated with ice-cold 100% acetone, following the procedure reported by Lichtenthaler (1987). The absorbance was detected at 470, 645 and 662 nm, and pigment content was expressed as μgcm−2.

Fruit antioxidant characterization

The effect of different light quality regimes on the antioxidant properties of ‘Microtom’ fruits was evaluated by collecting whole mature berries. Each assay was carried out on five fruits collected from five different plants, considering one berry as one replica. Fresh samples (0.250 g) were grounded in liquid nitrogen and the ascorbic acid (AsA) content, superoxide dismutase (SOD) and catalase (CAT) activities were determined as described in Arena et al. (2019).

The AsA concentration was evaluated with the Ascorbic Acid Assay Kit II (Sigma-Aldrich, St. Louis, MO, USA) based on the ferric reducing/antioxidant and ascorbic acid (FRASC) assay. Antioxidants contained in the sample are involved in reducing Fe3+ into Fe2+, resulting in a colored product. After the addition of ascorbate oxidase, any ascorbic acid is oxidized and quantified by measuring the absorbance at 593 nm with a spectrophotometer (UV-VIS Cary 100; Agilent Technologies, Santa Clara, CA, USA). The AsA concentration was determined using a standard curve and expressed in mg L−1, as reported in Costanzo et al. (2020).

The SOD Assay Kit (Sigma-Aldrich, St. Louis, MO, USA) was used to evaluate the SOD activity by measuring inhibition of the nitro blue tetrazolium (NBT) reduction into blue formazan. The absorbance of the blue color generated during the colourimetric reaction was read at 440 nm with a spectrophotometer (UV-VIS Cary 100; Agilent Technologies, Santa Clara, CA, USA). The volume of the sample that caused the 50% inhibition in blue formation was defined as a unit of SOD activity.

The CAT activity was assessed through the Catalase Assay Kit (Sigma-Aldrich, St. Louis, MO, USA). The colourimetric decomposition reaction of H2O2 into H2O and O2 was spectrophotometrically (UV-VIS Cary 100; Agilent Technologies, Santa Clara, CA, USA) followed by monitoring the decreasing absorbance at 520 nm. The amount of enzyme capable of decomposing 1 μmol of H2O2 per min at pH 7.0 and 25 °C was considered a CAT activity unit.

The total antioxidant capacity was assessed by the Ferric Reducing Antioxidant Power assay (FRAP) on samples (0.250 g) treated with methanol/water solution (60:40, v/v). As reported in George et al. (2004), samples were centrifuged at 20,817 g for 15 min at 4 °C, mixed with the FRAP reagents and incubated for 1 h in the dark. After the reaction, the absorbance was read at 593 nm. Then, the antioxidant capacity was calculated using a Trolox standard curve and expressed as μmol Trolox equivalents (μmol Trolox eq. g−1 FW).

The total polyphenols were quantified on samples (0.02 g) extracted with aqueous 80% methanol and subjected to the procedure described in Costanzo et al. (2020). The total polyphenol content was determined with a gallic acid standard curve and expressed as mg gallic acid equivalents (GAE) 100 g−1 FW.

Statistical analysis

Results were analyzed using SigmaPlot 12 software (Jandel Scientific, San Rafael, CA, USA). The effect of the different light quality treatments on the investigated parameters was assessed by applying a one-way analysis of variance (ANOVA). The Student–Newman–Keuls test was applied for all pairwise multiple comparison tests with a significance level of P < 0.05. The Kolmogorov–Smirnov and Shapiro–Wilk tests were performed to check for normality. Data are reported as mean values ± standard error (n = 5). All the data obtained for leaves and fruits were represented by two heatmapsto provide an immediate visual summary of information. The heatmaps were generated by means of the program ClustVis (https://biit.cs.ut.ee/clustvis/, accessed 31 January 2021). The clusters of rows and columns were based on Euclidean distance and average linkage. The numeric differences within each heatmap are indicated by a color scale: red scale from light todark indicated increasing values while blue scale decreasing values.

Results

Biometric measurements and leaf functional traits

The morphological parameters and leaf functional traits under the different light quality regimes were reported in Table 1. RGB and RB treatments reduced (P < 0.001) plant height (Table 1, Fig. 2) and increased (P = 0.002, P < 0.001) leaf number compared to FL light treatment. On the other hand, plants grown under the RGB regime developed the lowest number of flowers (P = 0.006, P = 0.011) and fruits (P = 0.001, P = 0.026) as well as a reduced (P < 0.001, P = 0.007) fruit total biomass than FL and RB plants. The growth under the three light regimes also induced a different partitioning of fresh biomass. More specifically, plants cultivated under RGB and RB light regimes invested more biomass into leaves (P < 0.001) and stem (P < 0.001, P = 0.015) (high ratio LB/EBand SB/EB) compared to FL plants. Conversely, FL and RB plants showed higher (P < 0.001) partitioning of biomass in fruits (high ratio FB/EB).

Table 1 Morphological parameters and leaf functional traits of ‘Microtom’ plants.

	Light quality regimes	
	FL	RB	RGB	
Morphological parameters				
Height (cm)	15.66 ± 0.658a	12.16 ± 0.556b	11.26 ± 0.370b	
Leaf number	22.60 ± 0.980c	37.60 ± 1.860a	30.0 ± 0.837b	
Flower number	50.00 ± 2.280a	47.00 ± 3.302a	36.60 ± 1.364b	
Fruit number	18.60 ± 1.939a	14.60 ± 0.872b	10.00 ± 0.632c	
Fruit weight (g)	34.78 ± 0.820 a	28.75 ± 3.400a	19.26 ± 0.692b	
SB/EB	0.234 ± 0.004c	0.250 ± 0.001b	0.272 ± 0.006a	
LB/EB	0.168 ± 0.006c	0.244 ± 0.010b	0.309 ± 0.003a	
FB/EB	0.598 ± 0.010a	0.506 ± 0.011b	0.420 ± 0.004c	
Leaf functional traits				
LA (cm2)	14.07 ± 0.494a	15.62 ± 0.588a	10.84 ± 0.433b	
SLA (cm2 g−1)	321.5 ± 11.25b	409.3 ± 8.900a	399.7 ± 9.824a	
RWC (%)	81.97 ± 0.736a	82.89 ± 0.850a	78.83 ± 1.080a	
LDMC (g g−1)	0.101 ± 0.003a	0.082 ± 0.001b	0.085 ± 0.003b	
Note:

Morphological parameters and leaf functional traits of S. lycopersicum L. cv. ‘Microtom’ plants cultivated under white fluorescent (FL), red-blue (RB) and red-green-blue (RGB) light regimes. Data are mean (n = 5) ± standard error. Different letters indicate statistically significant differences among light treatments (P < 0.05) according to one-way ANOVA. SB/EB, Stem biomass/epigeal biomass; LB/EB, leaf biomass/epigeal biomass; FB/EB, fruit biomass/epigeal biomass; LA, leaf area; SLA, specific leaf area; RWC, relative water content; LDMC, leaf dry matter content.

Figure 2 ‘Microtom’ plants grown under three different light quality regimes: fluorescent light (FL), red-green-blue (RGB) and red-blue (RB).

Representative view of Solanum lycopersicum L. ‘Microtom’ plants grown under three light quality regimes: white fluorescent light (FL), red-green-blue (RGB) and red-blue (RB). Scale bar = 2.5 cm.

Under RGB treatment, LA significantly decreased (P < 0.001) compared to FL and RB light regimes. An opposite behavior was observed for SLA and LDMC: FL plants showed a lower (P < 0.001) SLA and a higher (P < 0.001) LDMC compared to those grown under RGB and RB which exhibited comparable values. Lastly, RWC was not affected by different light quality treatments.

Gas exchange and chlorophyll fluorescence emission measurements

RGB light regime determined a significant increase (P < 0.001) of AN and gm compared to FL and RB treatments (Figs. 3A, 3C). Conversely, different light quality regimes did not affect gs (Fig. 3B). Consistent with AN, Vcmax was higher (P < 0.001) in RGB than FL and RB plants. The lowest value of Vcmax was measured in RB plants (Fig. 3D).

Figure 3 Gas exchanges of ‘Microtom’ plants under different light quality regimes.

(A) Net CO2 assimilation (AN), (B) stomatal conductance (gs), (C) mesophyll conductance (gm), (D) maximum rate of Rubisco carboxylation (Vcmax) in plants of Solanum lycopersicum L. ‘Microtom’ grown under three light quality regimes: white fluorescent light (FL), red-green-blue (RGB) and red-blue (RB). Data are expressed as mean ± standard error (n = 5). Different letters indicate statistically significant differences among light treatments (P < 0.05) according to one-way ANOVA.

The values of ΦPSII and Fv/Fm were lower in RB compared to RGB (P < 0.001, P < 0.001) and FL plants (P < 0.001, P = 0.001) (Figs. 4A, 4C). Consistently, RB plants also showed a higher (P < 0.001, P = 0.005) NPQ compared to RGB and FL plants (Fig. 4B). In particular, plants grown under the RGB regime exhibited the lowest (P < 0.004) NPQ.

Figure 4 PSII photochemistry of ‘Microtom’ plants under different light quality regimes.

(A) Quantum yield of PSII electron transport (ΦPSII), (B) non-photochemical quenching (NPQ), and (C) maximum PSII photochemical efficiency (Fv/Fm) in plants of Solanum lycopersicum L. ‘Microtom’ grown under three light quality regimes: white fluorescent light (FL), red-blue (RB) and red-green-blue (RGB). Data are expressed as mean ± standard error (n = 5). Different letters indicate statistically significant differences among light treatments (P < 0.05) according to one-way ANOVA.

Photosynthetic proteins and leaf pigments content

The plants cultivated under RB light significantly reduced (P = 0.034, P = 0.031) the content of D1 protein and Rubisco (P < 0.001, P = 0.011) compared to FL and RGB light regimes. No difference in D1 protein amount was found between FL and RGB plants. On the contrary, plants grown under RGB light showed the highest (P < 0.001) Rubisco amount among light treatments (Fig. 5).

Figure 5 Western blot and densitometric analysis of the D1 protein and Rubisco in ‘Microtom’ plants grown under white fluorescent (FL), red-blue (RB) and red-green-blue (RGB) light regimes.

Western Blot and densitometric analysis of the photosynthetic proteins D1 (A) and Rubisco (B) in Solanum lycopersicumL. ‘Microtom’ plants grown under three light quality regimes: white fluorescent light (FL), red-blue (RB) and red-green-blue (RGB). The bar diagrams represent pixel volumes expressed in arbitrary units of each band of D1 protein and Rubisco. Data are expressed as mean ± standard error (n = 3). Different letters indicate statistically significant differences among light regimes (P < 0.05) according to one-way ANOVA.

Compared to FL, plants grown under the RB regime significantly decreased the total chlorophyll and carotenoid content (P < 0.001, P = 0.027), while plants developed under the RGB regime only showed a lower chlorophyll concentration (P = 0.022) (Figs. 6A, 6B). An opposite trend was observed for Chla/b ratio, which resulted higher (P = 0.02) in RGB and even more in RB (P < 0.001) compared to FL plants (Fig. 6C).

Figure 6 Total chlorophylls (a+b), total carotenoids (x+c), and ratio between chlorophyll a and chlorophyll b (Chl a/b), in ‘Microtom’ plants grown under white fluorescent (FL), red-blue (RB) and red-green-blue (RGB) light regimes.

(A) Total chlorophylls (a+b), (B) total carotenoids (x+c), (C) ratio between chlorophyll a and chlorophyll b (Chl a/b), in Solanum lycopersicum L. ‘Microtom’ plants grown under three light quality regimes: white fluorescent light (FL), red-blue (RB) and red-green-blue (RGB). Data are expressed as mean ± standard error (n = 5). Different letters indicate statistically significant differences among light regimes (P < 0.05) according to one-way ANOVA.

Determination of antioxidants in fruits

The plant cultivation under RB light regime strongly affected the antioxidant properties of fruits. SOD and CAT activities, as well as the antioxidant capacity significantly increased (P < 0.001, P < 0.001, P < 0.001 respectively) in RB compared to FL and RGB fruits (Figs. 7A–7C). SOD and CAT activities did not differ between FL and RGB fruits, conversely to the antioxidant capacity, which was higher (P < 0.001) in RGB than FL fruits. Furthermore, the total polyphenol content also increased (P < 0.001) in RB compared to FL and RGB fruits, reaching a concentration about nine times higher than that found under the other two light regimes (Fig. 7E). On the other hand, the RB light regime did not promote the AsA content, which decreased (P < 0.001) in RB compared to FL and RGB fruits (Fig. 7D).

Figure 7 SOD and CAT activity, antioxidant capacity, ascorbic acid concentration, and total polyphenols in fruits of ‘Microtom’ plants grown under white fluorescent (FL), red-blue (RB) and red-green-blue (RGB) light regimes.

(A) SOD activity, (B) CAT activity, (C) antioxidant capacity, (D) ascorbic acid concentration, (E) total polyphenols in fruits of Solanum lycopersicum L. ‘Microtom’ plants grown under three light quality regimes: white fluorescent light (FL), red-blue (RB) and red-green-blue (RGB). Data are expressed as mean ± standard error (n = 5). Different letters indicate statistically significant differences among light regimes (P < 0.05) according to one-way ANOVA.

Heatmap analyses

An overview of the morphological, photosynthetic and functional traits of ‘Microtom’ plants in response to FL, RGB and RB light regimes is displayed in Fig. 8A.

Figure 8 Heatmaps showing plant morphological, physiological and biochemical traits and fruit characteristics of ‘Microtom’ under white fluorescent (FL), red-blue (RB) and red-green-blue (RGB) light regimes.

Cluster heatmap analysis summarizing plant morphological, physiological and biochemical traits (A) and fruit characteristics (B) of Solanum lycopersicum L. ‘Microtom’ plants cultivated under white fluorescent light (FL), red-blue (RB) and red-green-blue (RGB) light regimes. The color scale shows numeric differences within the data matrix: red and blue indicate increasing and decreasing values. Parameters are clustered in the rows; sample groups are clustered in the Light Quality factor columns.

The heatmap separated FL and RB from RGB plants, evidencing that an elevated amount of green wavelength in the light spectrum effectively promotes gas exchanges and carbon fixation, inducing higher values of AN, gs, gm, Vcmax, Rubisco content, and leaf biomass partitioning. Conversely, the FL regime grouped plants with more flowers and fruit biomass, higher photochemistry, photosynthetic pigment content and D1 protein amount. Finally, RB light regime clustered plants with high SLA, leaf number and chlorophyll a/b ratio.

Figure 8B summarizes the fruit traits, including the antioxidant properties. RB was separated from RGB and FL fruits. In particular, FL light regime induced higher fruit production and fruit weight. Conversely, the RB light regime clustered fruits with a higher antioxidant charge due to higher values of CAT and SOD activities, polyphenols and total antioxidant capacity.

Discussion

Our study showed that different light quality regimes (Fig. 1) strongly affect the photosynthetic and morphological traits in ‘Microtom’ plants and the antioxidant capacity of fruits, confirming that the modulation of the light spectrum is a valuable tool for controlling and selecting specific characters in this cultivar, especially in indoor environments.

Effect of different light quality regimes on photosynthetic and morphological traits of ‘Microtom’ plants

Regarding plant morphology (Table 1, Fig. 2), as previously reported by other authors, the growth under RB and RGB light quality regimes significantly reduced the stem elongation compared to the FL regime (Xiaoying et al., 2012; Arena et al., 2016; Dieleman et al., 2019; Izzo et al., 2020). A compact size characterizes the ‘Microtom’ cultivar, and the induction of further plant compactness may favour tomato growth in a high plant density condition or restricted volumes. Compared to FL, the higher intensityof blue wavelengths composing RB and RGB treatments may be responsible for the more compact size observed in these plants because blue wavelengths by inhibiting cell division and expansion act directly on plant morphogenesis, especially in the early stage of development (Dougher & Bugbee, 2004; Nanya et al., 2012; Izzo et al., 2020; Vitale et al., 2021).

The higher fruit number (Table 1) observed in FL than RB and RGB plants may depend on the far-red portion of the spectrum (2.9%) in this regime. It is noteworthy that plant morphogenesis is also controlled by phytochrome, regulated by the red/far-red ratio (Casal & Casal, 2000). According to other authors, the red/far-red ratio in FL regimes may have promoted stem extension, epigeal biomass, fruit yield and dry mass partitioning to fruits by increasing fruit sink strength in tomato plants (Ji et al., 2020; Kalaitzoglou et al., 2021).

The growth under RGB and RB regimes has induced biomass partitioning more in leaves than in fruits than in FL plants (Table 1). In the case of RGB plants, the investment toward photosynthetic structures led to a better photosynthetic performance than other light regimes.

Likely, the higher intensity of green wavelengths of RGB compared to RB regime may have favored photosynthesis. Indeed, the green component of the light spectrum, penetrating deeper into the leaf and reaching the lower cell layers than red or blue light, may have driven photosynthesis where the other wavelengths were limiting (Folta, 2005; Terashima et al., 2009; Smith, Mc Ausland & Murchie, 2017; Liu & van Iersel, 2021), and have favored photo-assimilate translocation in tomato leaves (Lanoue, Leonardos & Grodzinski, 2018). In FL plants, regardless of a high component of green, photosynthesis is lower than RGB, probably due to the lower mesophyll conductance.

The different light quality regimes also affected leaf functional traits (Table 1) indicating that leaf structural adjustments are required to allow plant acclimation to the surrounding light environment.

The growth under RGB and RB reduced the LDMC and increased the SLA compared to the FL regime indicating differences in the potential relative growth rate (Maizane & Shipley, 1999). Both SLA and LDMC are involved in the trade-off between quick biomass production (high SLA, low LDMC species) and efficient conservation of nutrients (low SLA, high LDMC species) (Poorter & De Jong, 1999); thus, the higher SLA in RGB and RB compared to FL plants suggests a more efficient growth strategy, under these specific light quality regimes.

Light-induced modifications of leaf structure strongly impact gas exchanges and photosynthetic carbon gain in ‘Microtom’ plants as determine changes in the resistances along the CO2 diffusion pathway inside leaves (Figs. 3A–3D) (Johkan et al., 2012; Arena et al., 2016; Vitale et al., 2020).

Despite similar values of SLA and LDMC, RGB and RB plants did not show the same photosynthetic efficiency. The higher AN in RGB compared to FL and RB plants was not due to the difference in stomatal conductance (gs) but increased mesophyll conductance (gm), indicating a reduced limitation to CO2 diffusion in mesophyll cells.

Thinner leaves, as well as less dense tissues in RGB plants (low LDMC), may reduce the limitations to the CO2 diffusion in the mesophyll (Niinemets et al., 2009; Tomás et al., 2013), leading to a higher amount of the CO2 available at the carboxylation sites, which, in turn, led to the significant increase of the maximum rate of Rubisco carboxylation (Vcmax) and net CO2 assimilation. This hypothesis is consistent with the highest level of Rubisco found in RGB plants (Fig. 5B).

The lack of significant differences in the stomatal conductance between RGB and RB plants suggested that AN decline in RB compared to RGB plants was not due to stomatal limitation but rather to other causes such as a decline of Rubisco activity. Indeed, it is noteworthy that a decreased capacity of ribulose-1,5-bisphosphate (RuBP) carboxylation or regeneration may be associated with lower photosynthetic performance (Onoda, Hikosaka & Hirose, 2005). Therefore, the low Rubisco expression may likely induce the Vcmax and AN drop observed in RB compared to FL and RGB plants (Figs. 3A, 3D). Consistent with our results, Miao et al. (2016) demonstrated in cucumber plants that the RB treatment (R:B 8:1) determined no change in gs but decreased Vcmax and photosynthesis compared to white fluorescent light.

The addition of green to red and blue wavelengths does not always produce positive effects on Rubisco expression and photosynthesis (Wang et al., 2009; Su et al., 2014); however, in our case, the more homogeneous light distribution within the leaf mesophyll (i.e., red and blue wavelengths on surface and green wavelength deeper in leaf parenchyma) may have induced stimulation of Rubisco synthesis (Terashima et al., 2009). Furthermore, similarly to Liu, Ren & Jeong (2019), adding green to red and blue wavelengths in our case also promoted D1 protein expression (Fig. 5A). The D1 levels were comparable to those found in FL plants, leading to a similar PSII photochemical efficiency (Fig. 4C). It is likely to suppose that in the RB regime, the high proportion of red wavelength did not favor photosynthesis because it negatively affected the Rubisco and D1 protein synthesis.

The growth under RB regime induced different partitioning of absorbed light energy within photosystems, promoting the heat dissipation processes instead of PSII photochemistry (Figs. 4A, 4B). Furthermore, the lowest content of total chlorophylls and carotenoids in RB compared to FL and RGB plants (Figs. 6A, 6B) also indicated a lower capability of light harvesting for these plants (Chen et al., 2014).

We cannot exclude that the down-regulation of photosynthetic pigments may represent in RB plants a safety strategy to reduce the light absorption, thus avoiding photodamages to PSII under limited photosynthetic activity. This hypothesis is supported by the increase of the chlorophyll a/b ratio that generally occurs in leaves exposed to higher light intensities (Kitajima & Hogan, 2003; Li et al., 2016). The increment of the Chl a/b ratio clearly indicates an adjustment of the light-harvesting system in RB plants, and more specifically, a reduction of Chl b mainly involved in the absorption of high-energy blue wavelengths (Wang et al., 2009). The Chl a/b ratios observed in this study are peculiar, as they deviate from the usual 3–4. However, many species of plantsshow values lower than the most commonly found. This may be considered a specific response to different light intensities or different light quality spectra, especially after long-term exposure to R, B and RB light (Kitajima & Hogan, 2003; Li et al., 2016; Zheng & Van Labeke, 2017).

The maintenance of the PSII activity is strictly related to the pigment concentration and the turnover of the D1 protein encoded by the psbA gene. The Fv/Fm decline in RB compared to FL and RGB plants (Fig. 4C) may indicate a slowdown of D1 turnover resulting from the imbalance between D1 degradation and replacement (Miao et al., 2016). As previously observed by Bian et al. (2018) in lettuce plants, in ‘Microtom’, the continuous RB light growth regime may have induced oxidative stress responsible for the downregulation of psbA expression and photosynthetic decline.

Effect of different light quality regimes on antioxidant properties of ‘Microtom’ fruits

The growth under different light qualities modified the antioxidant properties of tomato fruits (Fig. 7), evidencing that it is possible to obtain fruits richer in bioactive compounds for the human diet by manipulating the light spectrum. In particular, RB light strongly enhanced the antioxidant properties of ‘Microtom’ fruits, despite producing a lower number of berries per plant than FL (Table 1).

As in other crops, the total antioxidant capacity in tomato plants is due to compounds, such as carotenoids, ascorbic acid (AsA), vitamins, and polyphenols, which act as non-enzymatic defenses (Hasan et al., 2017; Ntagkas et al., 2019; Xie et al., 2019). AsA is considered one of the most potent scavengers in plant tissue and fruits (Racchi, 2013) and it has been recently demonstrated that light quantity and quality affect its production in tomato fruits (Ntagkas et al., 2019). Generally, the blue wavelengths of the light spectrum promote in detached tomato (Ntagkas et al., 2019) and strawberries fruits (Xu et al., 2018) an increase in AsA content compared to white fluorescent light, red or green wavelengths. Furthermore, the pure blue or dichromatic blue-red light also stimulated the AsA content in leafy vegetables (Ohashi-Kaneko et al., 2007; Li & Kubota, 2009; Ma et al., 2014). Our data indicate that the elevated antioxidant capacity of RB compared to FL and RGB fruits is not due to AsA but rather to the highest content of phenolic compounds (Figs. 7C–7E).

Our findings agree with previous studies on the same species, which demonstrated the stimulatory role exerted by RB light on the total polyphenols and antioxidant capacity (Xie et al., 2016). In particular, the wavelengths in the range of red, blue and UV-light strongly affect the accumulation of polyphenols, enhancing the antioxidant capacity and the reactive oxygen species (ROS) scavenging potential in tomato fruits (Castagna et al., 2014; Xie et al., 2016; Panjai et al., 2017). The higher antioxidant capacity induced by the RB treatment could be related to the cryptochromes, which induces the increase of flavonoids and lycopene (Giliberto et al., 2005). Specifically, cryptochromes are blue-light sensing photoreceptors whose activation can be inhibited by green light (Bouly et al., 2007). Therefore, the green fraction in FL and even more in the RGB regime may have offset the stimulatory effect of the RB wavelengths, determining a decrease in the antioxidant capacity of RGB and FL fruits (Fig. 7C).

The dichromatic RB regime also increased the scavenger enzymes SOD and CAT activity compared to FL and RGB fruits (Figs. 7A and 7B), likely due to the incidence of oxidative stress. Muñoz & Munné-Bosch (2018) reported that in different species, photooxidative stress could occur in fruits during the ripening. Thus, it cannot be excluded that the growth under the RB regime through a reduction of photosynthetic and photochemical activity may have induced oxidative stress in leaves and fruits activating the scavenging systems. In such circumstances, we hypothesized that during the scavenging of H2O2, the ascorbate peroxidase (APX) may have used the AsA as a co-factor (Racchi, 2013), contributing to its reduction in fruits of RB plants.

The heatmap (Fig. 8A) clustered FL and RB from RGB plants based on different physiological attributes, evidencing for RGB plants the best photosynthetic performance in terms of gas exchange and Rubisco amount. Conversely, FL regimes effectively promoted the reproductive structures (flower and fruit number). Concerning the fruits, the heatmap visualization (Fig. 8B) showed that the RB light regime greatly influenced the antioxidant production, except for AsA, suggesting the RB as the best light regime to guarantee fruits with a higher nutraceutical value, despite their low production under this treatment.

Conclusions

Overall results indicate that the photosynthetic apparatus of ‘Microtom’ grown under RGB treatments use light more efficiently than RB treatment. In fact, under the RGB growth regime, plants showed an improvement in photosynthetic performance, evidencing the important role of the green portion of the spectrum. Furthermore, the growth under RGB induced a more compact size and increased photochemical efficiency than FL and RB regimes. The increase of AN under RGB light treatment results from an improved mesophyll conductance due to changes in leaf structure and the up-regulation of Rubisco expression responsible for increasing maximum carboxylation efficiency in these plants.

However, despite the reduced photosynthetic performance, RB light regime stimulates the antioxidant production in ‘Microtom’ tomato fruits.

This study provides valuable information for developing appropriate light cultivation protocols through light manipulation to improve tomato plant productivity in controlled environments and the nutritional value of fruit quality, promoting the synthesis of antioxidants beneficial for the human diet.

Supplemental Information

Supplemental Information 1 Western blot 1.

Click here for additional data file.

Supplemental Information 2 Western blot 2.

Click here for additional data file.

Supplemental Information 3 Dataset.

Click here for additional data file.

Supplemental Information 4 The main outcomes of Microtom growth under light quality treatments.

Click here for additional data file.

Additional Information and Declarations

Competing Interests

Author Contributions

Data Availability

Carmen Arena is an Academic Editor for PeerJ.

Ermenegilda Vitale performed the experiments, analyzed the data, prepared figures and/or tables, authored or reviewed drafts of the article, and approved the final draft.

Violeta Velikova conceived and designed the experiments, performed the experiments, analyzed the data, prepared figures and/or tables, authored or reviewed drafts of the article, and approved the final draft.

Tsonko Tsonev performed the experiments, analyzed the data, prepared figures and/or tables, authored or reviewed drafts of the article, and approved the final draft.

Giulia Costanzo performed the experiments, prepared figures and/or tables, and approved the final draft.

Roberta Paradiso analyzed the data, authored or reviewed drafts of the article, and approved the final draft.

Carmen Arena conceived and designed the experiments, performed the experiments, analyzed the data, prepared figures and/or tables, authored or reviewed drafts of the article, and approved the final draft.

The following information was supplied regarding data availability:

The raw measurements are available in the Supplemental Files.

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
