# Peer review of "Manipulation of light quality is an effective tool to regulate photosynthetic capacity and fruit antioxidant properties of Solanum lycopersicum L. cv. ‘Microtom’ in a controlled environment"

_PeerJ, doi:10.7717/peerj.13677_

## Round 0.1 · original submission · Major Revisions

The reviewers raise concerns, which I share, about the classification of your light regimes, a subject that must be properly addressed. Additionally, the replication of your measurements should be more clearly presented in all cases; as an example, in L147-148 you refer to 5 leaves/plant/light treatment or 1leaf/plant – 5plants/light treatment?

The data of the photosynthetic pigments concentration and their ratios are peculiar; the chls/car is well above the usual 5-6 and the chl a/b well below the usual ~4. Please check the data and if the values are correct, give an explanation in the Results and Discussion sections.

Please consider dividing the Discussion section into paragraphs in relation to the Results.

Reviewer 1 ·

Basic reporting

Overall, the manuscript reads well. However there are some sentences which do need to be re-phrased in order to be published.

Experimental design

Please see section 4 for comments

Validity of the findings

Findings are valid and supported by the results presented. Please see section 4 for more comments.

Additional comments

One major concern I have is the way the authors classified the lighting treatments in figure 1. It seems that the authors have placed the FL treatment into many more narrow wavelength ranges. This has not been done with the RB and RGB treatments. While their name suggests they solely emit light in these wavelengths, this is clearly not the case in figure 1. Please revisit these classifications as it may change the results and arguments the authors are presenting in the discussion. I suggest either using only blue, green, red, and far-red as the wavelength classifications, or determining the percentage of the RB and RGB treatments which fall within the more narrow bands presented by the author.

Line 5: There is a space missing between controlled and environment. Potentially just a copying error?
Line 35: change to non-sustainable.
Line 58: remove the.
Line 87-93: Green light has also been shown to drive high rates of photosynthesis (similar to many other wavelengths) as well as support fruit filling through carbon export in tomato. I suggest this be mentioned in a way to strengthen the argument for green light. Suggest a reference to the following in this section as well: doi: 10.3389/fpls.2018.00756.
Line 108-110: This statement is quite awkward. I’m not sure what the authors mean by “finalized” in its current context. Please rephrase the sentence.
Figure 1: I suggest the authors either calculate all the percentages of wavelengths in all lights, or simplify it for the fluorescent light. For example, although the RB light is supposed to be purely red and blue, based on the graph, there is some violet in that spectrum. This is obviously a factor of the LED chip itself, but presenting the table in the way it is done is quite confusing. I suggest the authors use only blue, green, red, and far-red wavelength regions.
Plant material and growth conditions: Please add the number of plants which were place in each light treatment. Also, were there any replications studies performed? Technically, only one replicate seems to be performed with sub-samples (each plant) used for each measurements.
Line 132: When referring to epigeal plant biomass, I’m assuming the authors are speaking of all above ground biomass, however the next line they talk about leaf biomass/epigeal biomass. Please clarify is epigeal biomass is all above ground biomass or non-laminar biomass.
Line 151: Were the fully expanded leaves on separate plants.
Line 153-154: Please state the size of the chamber used for leaf photosynthetic measurements.
Line 155: What was the spectral quality of light used during the measurements?
Line 169: What was the illumination level for the steady-state fluorescence measurements?
Lines 201: Were the seeds of the tomatoes kept for the analysis or were they removed?
Line 254: This sentence is not needed. I suggest it be removed.
Line 259: I’m assuming the statement “Which showed comparable values” is referring to the FL and RB plants. However, it does not read this way and therefore should be changed.
Line 293: Remove indeed.
Heatmap analysis: Related to the wavelength ranges selected, the results may not be valid if correctly adjusted for each lighting source. Only FL light was split into defined wavelengths while RB and RGB was not. Looking at the spectral distribution, it seems that some green from the RGB treatment would be classified as cyan under the authors classification ranges. Perhaps this needs to be looked at again by the authors to ensure their results are conclusive and not a factor of how they classified wavelengths.
Line 327: Again, due to the authors wavelength ranges, these values for the amount of blue light may be different than reported.
Lines 392-394: This needs to be reworded as the beginning of the sentence does not read correctly.
Perhaps consider using subtitles in the discussion to break it into specific sections.

Reviewer 2 ·

Basic reporting

The manuscript by Vitale et al. describes the effect of different light spectra on the photosynthetic activity and fruit bioactive compound synthesis in Solanum lycopersicum L. cv. 'Microtom'. The article is well written with professional English used throughout. The authors have provided sufficient references and they have succeeded in presenting a professional article structure.
Nevertheless, throughout the discussion, the authors should add the number of the figure or table to which they refer to.

Experimental design

The research subject is within the Aims and Scope of the journal and the research question is well defined and meaningful. However, it is not stated how the authors fill an identified knowledge gap. The methods used are very well described with all the information to replicate.

Validity of the findings

All the data provided are robust and statistically sound and the conclusions are clear, well stated and linked to the original research question.

---

## Round 0.2 · Minor Revisions

The required changes were efficiently addressed in R1, but the following minor issues still need corrections.

L291: correct to FL
L297: correct to Determination of antioxidants in fruits
L339, 343, 364 and throughout the MS: separate the & from the names
L346, 353, 354, 355 and throughout the MS: separate et al from the names
L381: how this assumption/hypothesis is related or based to your results? Especially because you refer to “activity”. Is it related to the bibliographic information of the next sentence? In this case connect the paragraphs.
L379-381 and L439-440 and L464-467: please avoid the paragraphs of one sentence, unless there is an important reason for that; the sentence of these lines may be merged with the previous or the next paragraph.
L448-449: please rephrase to “whose activation can be inhibited by green light”
L456: consider changing the “determining” with a more suitable word to connect decreased photosynthetic efficiency and possible oxidative stress (e.g. through or causing).
L474: the better photosynthetic performance is stated two lines above (L472). Here the increased photochemical efficiency may be added.
Fig4: Please correct the FPSII to ΦPSII in the caption.
Fig 5A: the treatments should be presented with the same order in all figures, so please re-arrange the RGB and RB columns.

---

## Round 0.3 · accepted · Accept

The minor issues were addressed.